# Decreased Tongue Pressure Associated with Aging, Chewing and Swallowing Difficulties of Community-Dwelling Older Adults in Taiwan

**DOI:** 10.3390/jpm11070653

**Published:** 2021-07-11

**Authors:** Hsiu-Yueh Liu, Jen-Hao Chen, Kun-Jung Hsu, Ching-Teng Yao, Ping-Ho Chen, Szu-Yu Hsiao, Chun-Li Lin

**Affiliations:** 1Department of Oral Hygiene, College of Dental Medicine, Kaohsiung Medical University, Kaohsiung 80708, Taiwan; hyliu@kmu.edu.tw; 2Department of Medical Research, Kaohsiung Medical University Hospital, Kaohsiung Medical University, Kaohsiung 80708, Taiwan; 3School of Dentistry, College of Dental Medicine, Kaohsiung Medical University, Kaohsiung 80708, Taiwan; jehach@kmu.edu.tw (J.-H.C.); kjhsu@kmu.edu.tw (K.-J.H.); phchen@kmu.edu.tw (P.-H.C.); szyuhs@kmu.edu.tw (S.-Y.H.); 4Division of Prosthodontics, Department of Dentistry, Kaohsiung Medical University Hospital, Kaohsiung 80708, Taiwan; 5Department of Family Dentistry, Kaohsiung Medical University Hospital, Kaohsiung 80708, Taiwan; 6Department of Dentistry, Kaohsiung Municipal Ta-Tung Hospital, Kaohsiung 80145, Taiwan; 7Master Program of Long-Term Care in Aging, Kaohsiung Medical University, Kaohsiung 80708, Taiwan; angusyao@kmu.edu.tw; 8Cancer Center, Kaohsiung Medical University Hospital, Kaohsiung 80708, Taiwan; 9Division of Pediatric Dentistry and Special Needs Dentistry, Department of Dentistry, Kaohsiung Medical University Hospital, Kaohsiung 807377, Taiwan; 10Department of Biomedical Engineering, National Yang-Ming Chiao-Tung University, Hsinchu 30010, Taiwan

**Keywords:** tongue pressure, aging, epidemics, community, elderly

## Abstract

Personalized tongue pressure (TP) training focuses on improving swallowing. This study aims to establish the TP values of different age levels and compare changes between different swallowing status among community-dwelling elders. In this cross-sectional study, 1000 participants, aged 60 years old and above, were recruited from community care centers. All participants were classified into non chewing and/or swallowing difficulties (NCSD) and with chewing and/or swallowing difficulties (CSD) groups and their diseases and dieting status were recorded using a structured questionnaire. A disposable oral probe was used to measure TP by asking participants to compress it against the hard palate with maximum voluntary effort. Among 1000 elders, 63.10% had CSD and their TP (from 31.76 to 18.20 kPa) was lower than the NCSD group (from 33.56 to 24.51 kPa). Both groups showed the same tendency for TP decline with increasing age. Decline of TP makes CSD elderly have a poor appetite, eat a soft or liquid diet, and take longer to eat a meal (all *p* < 0.050). The secondary risk factor dominating TP decline for NCSD and CSD elders is having an education level less than primary school and an abnormal eating assessment, respectively. Our results demonstrated that TP decline has a significant relationship with age changes. Education level and an abnormal eating assessment score are closely associated with TP decline. A series of TP values can be used as a reference indicator of personalized medicine during the aging process among community-dwelling older adults.

## 1. Introduction

Taiwan’s accelerated rate of aging issues have become an imperative topic in the country’s long-term care policy and practice [1,2]. In response to the challenge of the rapidly growing older population, the new policy of long-term care, called LTC 2.0, was launched, planned, developed, and implemented in 2017 to prevent and delay the functional decline of the community-dwelling older population [3]. Aging has been confirmed to increase the decline of swallowing function by several studies [4,5,6,7,8,9]. Decreasing motor function in the lips, muscle strength and tongue pressure (TP), and loss of taste and smell with age increases and raises swallowing time, worsens swallowing performance, and impairs swallowing function [10]. A previous study in Taiwan found that 21.8% of the community’s elderly in Taiwan aged 65–95 years old tended to choke at least three times a week during eating, and 12.8% of the study population were assessed as having swallowing disorders [8]. Based on this proportion, it is reasonable to estimate that approximately 480,000 elders in Taiwan have swallowing disorders. Swallowing disorders and/or difficulties reduce the willingness and ability to eat food or drink liquids they once loved, causes nutritional imbalance, weight loss, dehydration, and also increases the risk of and mortality from aspiration pneumonia in the elderly [9].

The aging process sees diminished muscle mass, a decline in muscle strength, and causes frailty and sarcopenia. As for most muscle groups, muscle reduction also occurs in the tongue muscles. Tongue muscle loss reflects tongue muscle strength [11,12]. The performance of tongue strength is considered the main driving force for moving the bolus during the chewing and swallowing of food. Swallowing efficacy and safety may worsen due to muscle fatigue and weakness of the swallowing-related tongue muscles as a result of repetitive swallowing during a long meal time over 30 min [10,13]. Due to the development and progress of science, the strength of the tongue muscle can be measured via a mechanical pressure device exerted by the tongue against the hard palate. These devices provide an objective method of assessing tongue strength for community screening and physical research investigation [14,15]. A meta-analysis conducted by Adams et al. reported average TP decreased by about 8–9 kPa from young to old adults overall [16]. Another analysis further reported that mean TP value ranges of healthy individuals ascertained by JMS (JMS TPM-01/02, JMS Co., Ltd., Hiroshima, Japan) [17] were significantly higher in healthy individuals less than 60 years (37.5 kPa to 41.1 kPa) relative to those 60 years or older (29.4 kPa to 31.1 kPa) [18].

Over the past three decades, tongue muscle strength has been used as a useful indicator to evaluate the physically frail, the eating and swallowing ability of the elderly, to screen patients with or without penetration, aspiration, or residue during eating, and assist patients (such as those with stroke, Parkinson’s disease, and amyotrophic lateral sclerosis) with management targeted therapy goals or disease progression [13,14,15,19,20]. Personalized TP training is a new focus for improve swallowing. TP training or rehabilitation therapy at home is feasible, and it may become a new focus of personal precision medicine in the future. However, a large-scale TP database of older adults is currently unavailable in Taiwan. Moreover, the TP database may vary by country, subjects, and age groups. Before using TP to achieve personal swallowing precision medicine, it is necessary to establish a database of TP values among community-dwelling elderly. Therefore, this study aims to establish the TP values of different age levels among the community-dwelling elderly, and compare the TP changes between the elderly with different chewing and/or swallowing difficulties (CSD) status. These TP values of different age levels and CSD status may be used as an effective indicator for screening the changes of tongue function of the elderly and be an optimal treatment goal for patients with chewing and/or swallowing difficulties.

## 2. Methods

### 2.1. Study Design and Population

#### 2.1.1. Study Design

This observational cross-sectional study was carried out from October 2019 to November 2020. Community-dwelling older adults residing in Kaohsiung, Taiwan were enrolled in this study. The study adopted a multi-stage stratified cluster sampling design. The sampling probability is based on the probability proportional to size, with the community care centers as the sampling unit. First, Kaohsiung City has 38 districts stratified into seven clusters based on the urbanization stratification of townships in Taiwan [21]. We merged the first and second clusters into urban areas (8 districts with 92 care centers), third and fourth clusters into town areas (9 districts with 89 care centers), and fifth to seventh clusters into rural areas (21 districts with 112 care centers). Second, 38 community care centers were randomly selected from 293 community care centers based on the population probability of older adults (60 years old or above), which were 47.76% (19 care centers), 31.39% (11 care centers) and 20.65% (7 care centers) of the 3 district clusters, respectively. Third, we recruited the eligible older adults from each selected community care center. A total of 1000 participants were recruited from urban areas (*n* = 463), new town areas (*n* = 339), and rural areas (*n* = 198).

#### 2.1.2. Sample Size

The sample size calculation considered the population of 616,280 older adults living in the city. We wanted to identify the sample size for a larger population and used the Slovin or Yamane formulas [22]. This formula, based on Krejcie and Morgan’s recommendation [23], used 0.50 as an estimate of the population proportion to maximize variance, which will also produce the maximum sample size. With the population size of 616,280 older adults, at 95% confidence level and 5% margin of error, this resulted in our minimal sample size of 384 participants. We selected double the number of older adults necessary for research and considered 30% drop-out rate (as the rejection). Finally, the theoretical sample size in this designated study was at least 998 after original calculation. The actual sample size was adjusted to 1000 for further grouping and implementation. This study successfully recruited 1000 older adults (aged from 60 to 95 years old; 215 males, mean age of 74.75 ± 7.51 years; 785 females, mean age: 72.86 ± 7.58 years).

#### 2.1.3. Sample Selection and Randomization

The participants enrolled in this study (1) were equal to or older than 60 years; (2) had no craniofacial deformities or syndromes, no neuromuscular diseases, no history of head or neck cancer, had not undergone radiation therapy, and had no underlying neuromuscular diseases that were known as diseases that affect tongue strength. The exclusion criteria for the participants were (1) cognitively not fit for understanding and communication; (2) the coexistence of cognitive problems that affect language understanding, their inability to complete the tasks or the chance they may disrupt accurate TP measurement, and other problems that may impair the test.

#### 2.1.4. Ethical Approval

Ethical approval was obtained from the Human Experiment and Ethics Committees of Kaohsiung Medical University (Protocol number: KMUHIRB-F(I)-20190104). All procedures performed were in accordance with the ethical standards of the institutional and/or national research committee, and with the 1964 Helsinki Declaration and its later amendments or comparable ethical standards. Prior to oral examination and oral function evaluation, the purpose and content of the study protocol was thoroughly explained to the participants by the principal investigator. After agreeing to participate in the study, all participants signed a written informed consent.

### 2.2. Data Collection Methods

#### 2.2.1. Questionnaire Interview and Swallowing Assessment

A structured questionnaire was used to collect data on basic demographic characteristics (i.e., age, sex, education level), and diet status (i.e., type of diet and duration of meal time) referred to in the publication of Liu et al. [24]. Questionnaire data were collected by thoroughly trained interviewers using Mandarin or Taiwanese during face-to-face interviews in accordance with standard protocol. All interviewers participated in a 120-min training course on the standard process and data collection criteria to prevent information bias during interviews.

Swallowing difficulty screening used a self-administered questionnaire, called the Eating Assessment Tool (EAT-10) [25]. This self-administered tool is widely used to assess swallowing difficulty. According to the suggestion of EAT-10 normative data, a participant who scored 3 or more points on the EAT-10 was considered as having swallowing difficulty [25,26]. Each participant was also screened using the repetitive saliva swallowing test (RSST) within 30 s [27]. The RSST is a safe and simple screening test for functional swallowing difficulty. A participant who swallowed their saliva less than 3 times during 30 s of test was considered as having swallowing difficulty [27].

#### 2.2.2. Oral Examination

The oral examination of all study participants was performed by three well-trained and calibrated senior dentists according to the guidelines of the World Health Organization [28]. Intra-examiner reliability tests of tooth decay were carried out on these dentists during the oral data collection. The kappa coefficient was 0.81 for intra-examiner agreement. The participant’s oral health examinations were performed in a group activity room in the care centers between 9.00 a.m. and 11.00 a.m. under natural light using disposable dental mirrors and CPI explorers, without radiographs.

Information was collected on the kind of denture and the condition of natural teeth, which included the caries status, location, and number. The dentist counted the number of functional natural teeth (FNT), fixed artificial teeth (FAT), removable artificial teeth (RAT), and functional teeth (FT) [29,30]. All examinations excluded third molars. Remaining natural teeth, teeth that were sound, decayed, filled, or filled but decayed were regarded as FT. Teeth with grade III mobility, residual roots, or extensive crown destruction (i.e., more than three-fourths of the clinical crown destroyed) were excluded. FAT was defined as fixed artificial teeth, including abutment teeth, pontics, and implant-supported prostheses. FAT with grade III mobility were excluded. RAT was regarded as artificial teeth on removable dentures being worn during the dental examination. FT included FNT, FAT, and RAT. The total number of FT ranged from 0 to 28. The participants with more teeth was helpful for assessment of TP [11,26,31]. According to the number of FT, participants were classified into 0–9, 10–19, and ≥20 teeth groups to analyze the relationship between FT and TP [11].

#### 2.2.3. Tongue Pressure Measurement

A wireless TP measurement device (OCPT-168, Shi-Heng Technology Co., Ltd., Taipei, Taiwan) was used to assess TP [32]. The device is a personal use device which consists of a disposable position mouthpiece and a pressure transducer (Figure 1a). During the test, the participant sits with foot support, keeps head upright, and eyes on a horizontal target (Figure 1b). A mouthpiece is used for each participant. The TP mouthpiece is placed behind the central incisor along the central groove of the tongue blade. The participant is asked to press their tongue against the hard palate as hard as possible for 5 s and a maximum TP is obtained on the digital screen of the device. Each participant was tested 3 times with 30-s rest interval in between. The obtained maximum value was recorded as maximum TP of each participant. For the data analysis, participants’ TP less than the median value (27 kPa) was considered as decreased TP.

### 2.3. Statistical Analysis

In order to clarify the association between TP and CSD, the participants were divided into non CSD (NCSD) and CSD groups. If the participants had fewer than 20 FNT an RSST equal or less than three times, or an EAT-10 score equal or more than 3 or self-reported as having swallowing difficulty [25,26], they were classified into the CSD group. The participants of the NCSD group were the healthy older adults, who had no chewing or swallowing difficulties. Further to this, the participants in both groups (NCSD and CSD) were categorized into 6 age levels (60–64 years, 65–69 years, 70–74 years, 75–79 years, 80–84 years, and 85 years or above) to identify age-related changes in TP.

The participants’ demographic characteristics, questionnaire data, and TP were entered into Microsoft Excel (Microsoft, Redmond, WA, USA). IBM SPSS Statistics 20.0 (IBM, Armonk, NY, USA), which was then used to perform descriptive and inferential statistical analysis. Demographic characteristics were expressed as numbers and percentages, and TP was presented as means and standard deviations. Chi-square test was used to compare the demographic characteristics between NCSD and CSD groups. The *t*-test and ANOVA were used to compare the TP value between or among different variable groups. Pearson correlation analysis was used to measure the correlation between age and TP in both groups. To determine factors involved in TP, multivariate linear regression analysis was used. In addition, in order to obtain the best explanatory models of age changes on decreased TP, univariate and multivariate logistic regression analysis was used. In all analyses, the significance level was set at 5% and the confidence interval was 95%.

## 3. Results

There were 63.10% of older adults who had CSD. The distribution of CSD had a statistically significant increase with age, lower education level, more chronic diseases, fewer teeth, lower RSST times, and a higher score of EAT-10 (all *p* < 0.001) (Table 1).

Both males and females aged 60–64 years old had the highest average TP compared with other age groups, as shown in Figure 2. The average TP was 30.63 ± 11.77 kPa in the NCSD group, and this was statistically significantly higher than the CSD group (25.17 ± 13.61 kPa) (*p* < 0.001) (Table 2). The NSCD and CSD participants whose education level was university or above, had the highest TP (34.67 kPa and 29.37 kPa) compared with those who had other education levels (all *p* = 0.001) (Figure 3). The TP value of CSD had a positive tendency with the number of functional teeth, but did not show any statistically significant difference. In the NCSD and CSD groups, TP statistically significantly decreased with age increase (all *p* < 0.001). The scatter plots and bar charts revealed there was a negative significant relationship between TP and age in both the NCSD and CSD groups (all *p* < 0.001) (Figure 4).

In order to better understand the participants, we further compared the dietary habits between the NCSD and CSD groups, as shown in Table 3. Higher TP values of CSD participants had a statistically significant benefit with their independent ability to eat, various types of diet, and shorter duration of meal time (all *p* < 0.050) while eating a meal. As TP decreased, CSD participants obviously felt that they needed to take longer to eat a meal than before (*p* = 0.006).

Multiple linear regression models were performed to clarify the risk factors of decreased TP among different target groups (Table 4). In the three regression models, we found the only common risk factor in the total population and for NCSD and CSD groups was age level. Other risk factors which affected TP were lower education level, fewer FT and a higher EAT-10 score.

Finally, we reconfirmed the most important risk factor for decreased TP among older adults was age level through multiple logistic regression models (Table 5). Whether the older adults had CSD or not, the odds ratio of decreased TP which was 1.54 (AOR = 1.54, 95% CI: 0.98–2.43, *p* = 0.062) in the 65–69 years old group increased to 6.61 (AOR = 6.61, 95% CI: 3.58–12.57, *p* < 0.001) in the 85 years old or above group compared with the 60–65 years old group.

## 4. Discussion

Personalized TP training or rehabilitation therapy is a new measure to improve swallowing with a basis of customized design. In the present study, we successfully established a large amount of TP data in NCSD (healthy) and CSD adults over the age of 60 by a quantitative method. The results of this study show evidence that TP declines are in parallel for NSCD and CSD adults at every 5-year increase in age. But the TP of CSD adults saw an accelerated decline from 60 years old (31.76 kPa) until 85 years or older (18.20 kPa) than NCSD adults (33.56 kPa and 24.51 kPa). Furthermore, this study established the standard and normal reference ranges of TP values at 60 years (31.47 kPa), 70 years (26.22 kPa), and 80 years and above (20.83 kPa) of overall participants. If an elder’s TP value is less than 30 kPa, 25 kPa, and 20 kPa, respectively, he or she may have CSD. These three TP values can also be used as reference indicators for community screening, clinical diagnosis, and treatment effectiveness of CSD. In the future, the elderly can refer to the TP values currently found under parameters such as age, gender, education level, CSD status, etc., to set personalized training goals for rehabilitation.

Besides the 60–64 and 75–79 years old groups, there was no difference in TP values between genders of most age groups in the present study, as seen in several previous studies [6,17,33,34]. The TP values in males was rather gradual with a slightly higher value (within 2.5 kPa) than in females in most age groups. A research reviewed numerous studies and concluded that the impact of TP in healthy older adults between different genders only exists in the participants who are less than 60 years old [18]. From the present study results, we can clearly see that the TP of male older adults at 60–64 years and 75–79 years are significantly higher than females. This may because the height of a male’s anterior tongue have already decreases by their sixties [34]. Then, a man may experience greater loss of total muscle mass at this time, accompanied with a rapid decline in muscle strength at age 75 or older, until they finally reach the same level of pressure as females [5,17].

This study clearly demonstrates that TP decreased with natural aging in older adults. Past studies showed that mean TP values in healthy individuals at an age less than 60 years was 39.3 kPa, which was higher than those at age 60 years or above (30.3 kPa) [18]. Furthermore, we compared our TP values with several studies of similar age composition. These studies included the participants’ age between 60 years to 95 years and the mean age of participants was between 70–75 years. We found that our mean TP conducted by TP wireless application (30.63 kPa) was similar with previous studies which were conducted using JMS (29.5 kPa to 31.8 kPa) [15,17,18,35,36]. This result confirmed that our TP data have high reference value. Hara et al. reported that the TP values of healthy people in their forties, fifties, sixties, seventies, and eighties were 36.48 kPa, 34.59 kPa, 33.16 kPa, 29.92 kPa, and 23.60 kPa, respectively [37]. From the above data, we observed that the TP values in 60–64 year olds was lower than those less than 60 years old. We also observed that our TP value for 60–64 year olds (33.56 kPa) was very close to the relevant study of Hara et al. (33.16 kPa). We confirmed that this study’s results provide quantitative evidence of the age-related changes in TP during the aging process.

CSD is another risk factor of rapid TP decline under natural aging. From 75 years of age, TP continued to rapidly decrease with larger differences with increasing age between the NCSD and CSD groups until the later stage of aging. We observed that the TP decline rate from 60–64 years old to 80 years old or older of CSD (42.70%) participants were 1.5 times that of NCSD participants (26.97%). Wang et al. reported that swallowing function decreases significantly above the age of 75 years old in Taiwan [8], supporting the results of the present study. However, the former TP studies showed no evidence of independent differences in CSD attributable to age due to only a small sample size of the elderly group evaluated or a lack of clear health conditions of the participants [6,18,38]. We collected chronic disease, teeth number, swallowing, and eating status of each participant through a large sample size and excluded the relationship between diseases and TP during data analysis. Unquestionably, CSD accelerates the decline rate of TP. The study provides a supplement to explore the interaction between age and swallowing difficulty in TP.

This study observed that decreased TP strongly affects the eating status of CSD. The elderly who have swallowing difficulties commonly encounter swallowing muscle fatigue, longer mealtimes, low food consumption, and malnutrition, supporting the results of the present study [36,39]. A series of negative effects during the eating process may originate from weak TP. A low TP makes normal swallowing hard to achieve. As noted in previous studies, the percentage of maximum TP used during swallowing in older adults was 53.8%, which was higher than young adults at 38.8%. This could be regarded as an indispensable compensatory mechanism to complete safe swallowing [7] because posterior TP is needed to generate propulsion to transfer solid bolus into the pharynx [12]. Based on the current findings, order adults with CSD had lower TP than those without CSD and a decreasing TP is linked to CSD. This finding reveals that a lack of adequate tongue muscle strength is one of the eating disorders in older adults with CSD.

There are some limitations that need to be noted for the current study. First, we cannot well observe the impact of disease on TP because those who were actively willing to participate in this study may tend to be healthier than those who did not participate. Due to an inability to reach the complete population of elderly, the results of the current study may be overestimated. Second, the presence of fewer males than females may elicit selection bias. However, the number of each age group in this study were larger than in previous studies, and we believe that the results of the study can compensate for part of the sample bias.

## 5. Conclusions

TP decline is prevalent among community-dwelling older adults, and aging is the major factor in TP decrease in parallel with ordinary and CSD adults. Older adults with CSD have an accelerated decrease in their TP, especially when they are 75 years old or above. This study successfully established a series of TP values in NCSD (from 33.56 to 24.51 kPa) and CSD (from 31.76 to 18.20 kPa) in Taiwan and suggests that TP can be used as a reference indicator of aging progression among community-dwelling older adults. The authors have confirmed that all authors meet the Journal of Personalized Medicine criteria for authorship credit, as follows: (1) substantial contributions to conception and de-sign, acquisition of data, or analysis and interpretation of data; (2) drafting the article or revising it critically for important intellectual content; and (3) final approval of the version to be published. 

## Figures and Tables

**Figure 1 jpm-11-00653-f001:**
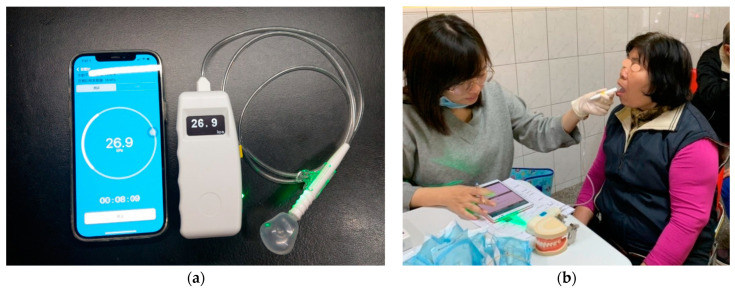
(**a**) A tongue pressure measurement device with wireless mobile application control function and a disposable oral positioning mouthpiece. (**b**) The tongue pressure device and its usage in the measurement of tongue pressure.

**Figure 2 jpm-11-00653-f002:**
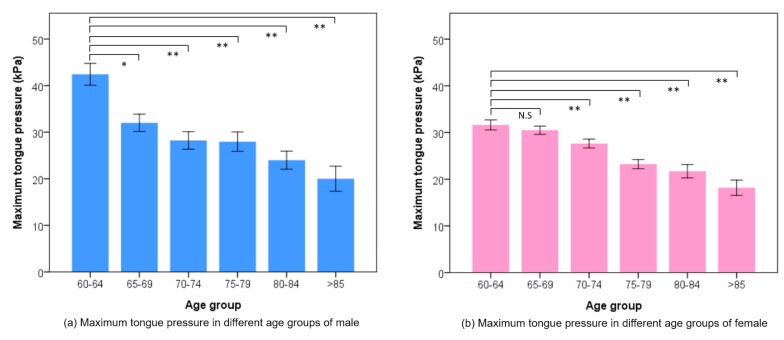
The mean maximum tongue pressure of (**a**) male and (**b**) females among age groups. Each bar showed the mean value and standard error. * *p* < 0.05, ** *p* < 0.01. N.S.: non-significance.

**Figure 3 jpm-11-00653-f003:**
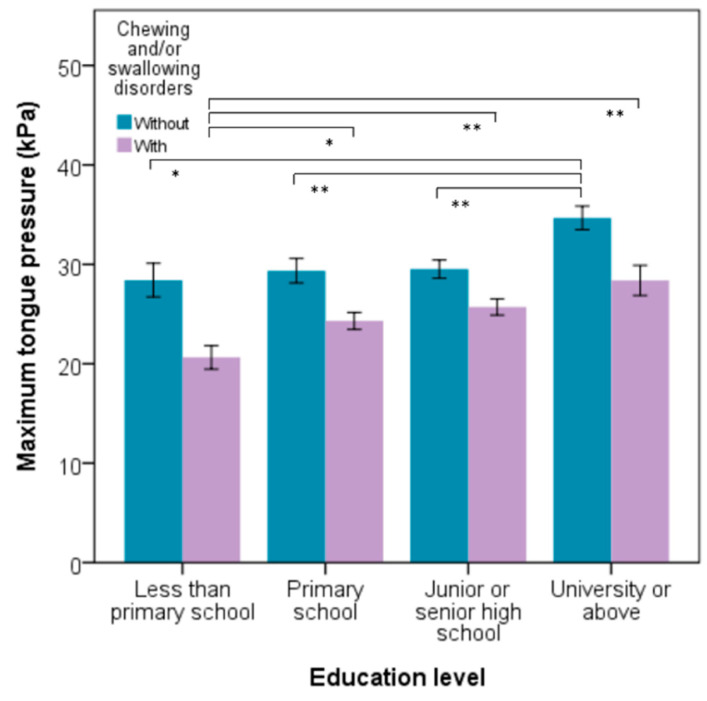
Comparison of maximum tongue pressure of participants without/with chewing and/or swallowing difficulties by different education levels. Each bar shows the mean value and standard error. * *p* < 0.05, ** *p* < 0.01.

**Figure 4 jpm-11-00653-f004:**
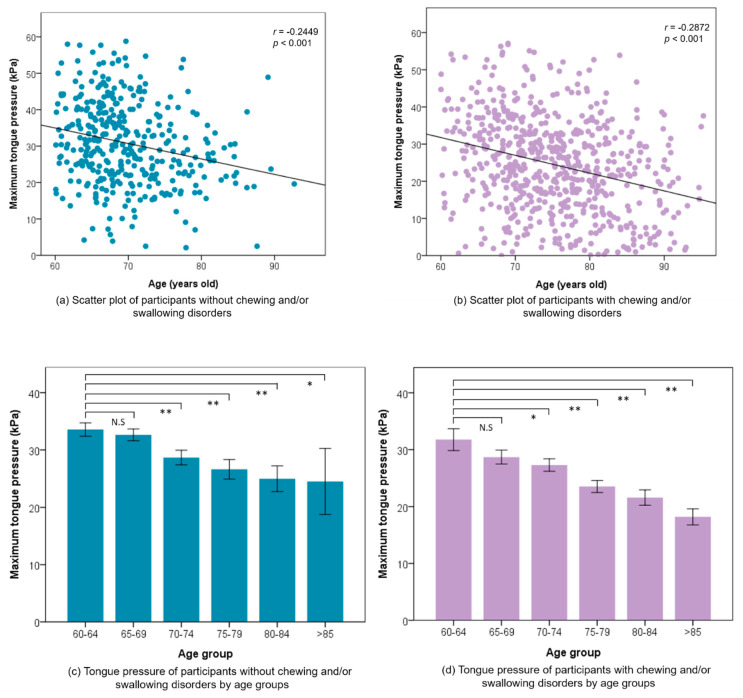
Correlation between maximum tongue pressure and age of both participants (**a**) without and (**b**) with chewing and/or swallowing difficulties. Tongue pressure of participants (**c**) without and (**d**) with chewing and/or swallowing difficulties by age groups. Each bar shows the mean value and standard error. * *p* < 0.05, ** *p* < 0.01. N.S.: non-significance.

**Table 1 jpm-11-00653-t001:** General characteristic of participants.

Variable	Total		Without CSD		With CSD		*p*-Value
	*n*	(%)	*n*	(%)	*n*	(%)	
Total	1000		369	(36.90)	631	(63.10)	
Gender							
Male	215	(21.50)	79	(21.41)	136	(21.55)	0.9574
Female	785	(78.50)	290	(78.59)	495	(78.45)	
Age group							
60–64 yrs	132	(13.20)	80	(21.68)	52	(8.24)	<0.001
65–69 yrs	261	(26.10)	137	(37.13)	124	(19.65)	
70–74 yrs	222	(22.20)	75	(20.32)	147	(23.30)	
75–79 yrs	181	(18.10)	44	(11.92)	137	(21.71)	
80–84 yrs	119	(11.90)	26	(7.05)	93	(14.74)	
≥85 yrs	85	(8.50)	7	(1.90)	78	(12.36)	
Education level							
Less than primary school	145	(14.50)	27	(7.32)	118	(18.70)	<0.001
Primary school	354	(35.40)	95	(25.74)	259	(41.04)	
Junior or senior high school	334	(33.40)	148	(40.11)	186	(29.48)	
University or above	167	(16.70)	99	(26.83)	68	(10.78)	
Marital status							
Married	345	(34.50)	101	(27.37)	244	(38.67)	<0.001
Single	655	(65.50)	268	(72.63)	387	(61.33)	
Discretionary income							
Sufficient	921	(92.10)	353	(95.66)	568	(90.02)	0.0014
Insufficient	79	(7.90)	16	(4.34)	63	(9.98)	
Chronic disease							
No	322	(32.20)	151	(40.92)	171	(27.10)	<0.001
Yes	678	(67.80)	218	(59.08)	460	(72.90)	
Functional natural teeth							
0–9 teeth	236	(23.60)			236	(37.40)	<0.001
10–19 teeth	186	(18.60)			186	(29.48)	
≥20 teeth	578	(57.80)	369	(100.00)	209	(33.12)	
Functional teeth							
0–9 teeth	19	(1.90)			19	(3.01)	<0.001
10–19 teeth	52	(5.20)			52	(8.24)	
≥20 teeth	929	(92.90)	369	(100.00)	560	(88.75)	
EAT-10 score							
<3	849	(84.90)	369	(100.00)	480	(76.07)	<0.001
≥3	151	(15.10)			151	(23.93)	
RSST times							
<3	200	(20.00)			200	(31.70)	<0.001
≥3	800	(80.00)	369	(100.00)	431	(68.30)	

yrs: years old.

**Table 2 jpm-11-00653-t002:** Tongue pressure of participants.

Variable	*N*	Total		*p*-Value	Without CSD		*p*-Value	With CSD		*p*-Value
		Mean	SD		Mean	SD		Mean	SD	
Total	1000	27.18	13.22	0.125	30.63	11.77		25.17	13.61	<0.001
TP range (Q1–Q3) ^a^		(18.10–36.00)			(22.25–39.25)			(15.40–33.60)		
Gender										
Male	215	28.41	13.71	0.125	32.54	12.03	0.104	26.01	14.10	0.416
Female	785	26.85	13.07		30.11	11.67		24.94	13.47	
Age group										
60–64 yrs	132	32.85	11.93	<0.001	33.56	10.53	<0.001	31.76	13.84	<0.001
65–69 yrs	261	30.77	12.89		32.64	12.10		28.71	13.46	
70–74 yrs	222	27.78	12.57		28.69	10.91		27.32	13.35	
75–79 yrs	181	24.30	12.20		26.65	11.32		23.55	12.42	
80–84 yrs	119	22.34	12.76		25.00	11.47		21.60	13.05	
≥85 yrs	85	18.72	12.77		24.51	15.23		18.20	12.51	
Education level										
Less than primary school	145	22.64	13.17	<0.001	28.41	8.86	0.001	21.32	13.66	0.001
Primary school	354	26.19	14.04		29.03	12.29		25.14	14.52	
Junior or senior high school	334	27.55	11.60		29.36	11.35		26.10	11.63	
University or above	167	32.51	12.75		34.67	11.78		29.37	13.54	
Marital status										
Married	345	25.48	13.43	0.003	28.76	11.95	0.062	24.12	13.79	0.124
Single	655	28.08	13.03		31.33	11.65		25.83	13.46	
Discretionary income										
Sufficient	921	27.28	13.10	0.423	30.63	11.75	0.980	25.20	13.47	0.848
Insufficient	79	26.04	14.58		30.70	12.61		24.86	14.90	
Chronic disease										
No	322	27.17	12.64	0.988	30.09	11.40	0.467	24.60	13.14	0.520
Yes	678	27.19	13.49		31.00	12.03		25.38	13.78	
Functional natural teeth										
0–9 teeth	236	23.19	14.98	<0.001				23.19	14.98	0.018
10–19 teeth	186	26.32	13.58					26.32	13.58	
≥20 teeth	578	29.09	11.90		30.63	11.77		26.38	11.66	
Functional teeth										
0–9 teeth	19	20.03	14.13	0.022				20.03	14.13	0.237
10–19 teeth	52	24.81	13.53					24.81	13.53	
≥20 teeth	929	27.46	13.14		30.63	11.77		25.38	13.58	
EAT-10 score										
<3	849	27.94	12.95	<0.001	30.63	11.77		25.87	13.44	0.020
≥3	151	22.93	13.93					22.93	13.93	
RSST times										
<3	200	27.94	12.95	<0.001				24.20	14.77	0.244
≥3	800	22.93	13.93		30.63	11.77		25.62	13.02	

yrs: years old; CSD: chewing or swallowing difficulties; ^a^: Data are expressed as interquartile range.

**Table 3 jpm-11-00653-t003:** Cross table of diet status and tongue pressure of participants by their eating status.

Variable	*N*	Total		*p*-Value	*N*	Without CSD		*p*-Value	*N*	With CSD		*p*-Value
		Mean	SD			Mean	SD			Mean	SD	
Ability to eat												
Completely by themselves	985	27.31	13.19	0.017	365	30.63	11.82	0.942	62	25.35	13.57	0.013
Partially assisted by parents/caregivers	15	19.09	12.67		4	30.20	7.43		11	15.05	11.87	
Appetite												
Good	808	27.93	13.19	<0.001	321	31.30	11.63	0.014	487	25.71	13.68	0.080
Satisfactory	163	24.69	12.96		45	26.40	12.12		118	24.04	13.26	
Poor	29	20.37	12.18		3	21.83	3.64		26	20.20	12.84	
Types of diet												
General diet, rice	914	28.18	12.98	<0.001	363	30.79	11.70	0.112	551	26.46	13.49	<0.001
Soft diet, rice porridge	62	17.15	11.06		5	20.42	14.57		57	16.86	10.82	
Liquid diet	24	15.01	10.68		1	21.90	.		23	14.71	10.82	
Duration of meal time												
<16 min	636	28.06	13.40	0.006	238	30.94	11.99	0.493	398	26.33	13.91	0.005
≥16 min	364	25.66	12.77		131	30.06	11.38		233	23.18	12.86	
Feel taking longer time to eat a meal than before												
Obviously	38	20.03	12.50	<0.001	6	29.58	9.01	0.883	32	18.23	12.35	0.006
Slightly	173	25.46	13.79		37	29.82	12.44		136	24.27	13.94	
No	789	27.91	13.00		326	30.74	11.76		463	25.91	13.46	

CSD: chewing and/or swallowing difficulty.

**Table 4 jpm-11-00653-t004:** Factors affecting tongue pressure of participants.

Variable	Estimate	SE	*t* Ratio	*p*-Value	95%CI		*R* ^2^
					(Lower,	Upper)	
Total participants							
Gender							
Female (vs. Male)	−3.01	0.97	−3.11	0.002	(−4.90,	−1.11)	0.127
Age group							
65–69 yrs (vs. 60–64 yrs)	−2.28	1.33	−1.72	0.086	(−4.88,	0.32)	
70–74 yrs (vs. 60–64 yrs)	−5.42	1.37	−3.96	<0.001	(−8.10,	−2.73)	
75–79 yrs (vs. 60–64 yrs)	−8.53	1.43	−5.99	<0.001	(−11.33,	−5.74)	
80–84 yrs (vs. 60–64 yrs)	−10.49	1.58	−6.64	<0.001	(−13.58,	−7.39)	
≥85 yrs (vs. 60–64 yrs)	−13.28	1.77	−7.50	<0.001	(−16.75,	−9.81)	
Functional natural teeth (Teeth)	0.20	0.09	2.22	0.027	(0.02,	0.38)	
EAT-10 (Score)	−0.43	0.12	−3.52	0.001	(−0.67,	−0.19)	
Participants without CSD							
Gender							
Female (vs. Male)	−2.53	1.57	−1.61	0.11	(−5.61,	0.55)	0.101
Age group							
65–69 yrs (vs. 60–64 yrs)	−1.34	1.61	−0.83	0.41	(−4.49,	1.82)	
70–74 yrs (vs. 60–64 yrs)	−5.39	1.91	−2.83	0.00	(−9.12,	−1.66)	
75–79 yrs (vs. 60–64 yrs)	−7.11	2.20	−3.23	0.00	(−11.43,	−2.79)	
80–84 yrs (vs. 60–64 yrs)	−8.72	2.68	−3.26	0.00	(−13.96,	−3.47)	
≥85 yrs (vs. 60–64 yrs)	−9.25	4.52	−2.05	0.04	(−18.11,	−0.39)	
Education level							
Less than primary school (vs. University or above)	−4.32	1.52	−2.84	0.00	(−7.30,	−1.33)	
Primary school (vs. University or above)	−3.04	1.76	−1.73	0.09	(−6.48,	0.41)	
Junior or senior high school (vs. University or above)	−2.46	2.66	−0.93	0.35	(−7.68,	2.75)	
Participants with CSD							
Gender							
Female (vs. Male)	−2.22	1.27	−1.75	0.081	(−4.70,	0.27)	0.100
Age group							
65–69 yrs (vs. 60–64 yrs)	−3.13	2.14	−1.46	0.145	(−7.34,	1.07)	
70–74 yrs (vs. 60–64 yrs)	−4.66	2.10	−2.22	0.027	(−8.77,	−0.55)	
75–79 yrs (vs. 60–64 yrs)	−8.40	2.12	−3.97	<0.001	(−12.55,	−4.25)	
80–84 yrs (vs. 60–64 yrs)	−10.26	2.26	−4.55	<0.001	(−14.69,	−5.84)	
≥85 yrs (vs. 60–64 yrs)	−13.02	2.35	−5.53	<0.001	(−17.63,	−8.40)	
EAT-10 (Score)	−0.37	0.13	−2.80	0.005	(−0.63,	−0.11)	

yrs: years old; CSD: chewing and/or swallowing difficulties; CI: confidence interval. Adjusted gender.

**Table 5 jpm-11-00653-t005:** Factor of decreased tongue pressure of participants

Variable		Total				Total				Without CSD				With CSD		
	COR ^a^	95%CI		*p*-Value	AOR ^b^	95%CI		*p*-Value	AOR ^b^	95%CI		*p*-Value	AOR ^b^	95%CI		*p*-Value
		(Lower,	Upper)			(Lower,	Upper)			(Lower,	Upper)			(Lower,	Upper)	
Gender																
Female (vs. Male)	1.18	(0.87,	1.60)	0.277	1.44	(1.04,	1.98)	0.026	1.82	(1.06,	3.21)	0.032	1.25	(0.84,	1.87)	0.265
Age group																
65–69 yrs (vs. 60–64 yrs)	0.56	(0.42,	0.74)	<0.001	1.54	(0.98,	2.43)	0.062	1.40	(0.77,	2.61)	0.276	1.71	(0.88,	3.44)	0.121
70–74 yrs (vs. 60–64 yrs)	0.92	(0.68,	1.24)	0.588	2.30	(1.46,	3.66)	<0.001	2.97	(1.51,	5.97)	0.002	1.90	(0.99,	3.75)	0.059
75–79 yrs (vs. 60–64 yrs)	1.77	(1.20,	2.64)	0.004	3.95	(2.33,	6.79)	<0.001	3.36	(1.34,	8.60)	0.010	3.65	(1.88,	7.33)	<0.001
80–84 yrs (vs. 60–64 yrs)	1.87	(1.35,	2.62)	<0.001	4.01	(2.48,	6.55)	<0.001	4.28	(1.97,	9.59)	<0.001	3.89	(1.91,	8.16)	<0.001
≥85 yrs (vs. 60–64 yrs)	3.16	(1.94,	5.33)	<0.001	6.61	(3.58,	12.57)	<0.001	7.54	(1.48,	56.18)	0.022	6.21	(2.91,	13.80)	<0.001
Education level																
Less than primary school (vs. University or above)	1.73	(1.21,	2.50)	0.003	1.55	(1.04,	2.32)	0.034	1.83	(1.03,	3.29)	0.040				
Primary school (vs. University or above)	1.34	(1.03,	1.74)	0.027	1.56	(1.03,	2.36)	0.036	1.58	(0.82,	3.05)	0.171				
Junior or senior high school (vs. University or above)	0.88	(0.67,	1.14)	0.326	1.53	(0.90,	2.60)	0.115	1.43	(0.54,	3.77)	0.471				
Functional natural teeth <20																
Yes (vs. No)	1.86	(1.34,	2.60)	<0.001												
EAT-10 score ≥3																
Yes (vs. No)	1.88	(1.32,	2.69)	0.001	1.49	(1.03,	2.19)	0.037					1.37	(0.93,	2.03)	0.118

yrs: years old; CSD: chewing and/or swallowing difficulties; CI: confidence interval. ^a^ COR = Crude Odds Ratio. Data analysis by simple logistic regression model. ^b^ AOR = Adjusted Odds Ratio. Data analysis by multiple logistic regression models. Adjusted gender.

## Data Availability

Not applicable.

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
