# Peer review of "Decreased Tongue Pressure Associated with Aging, Chewing and Swallowing Difficulties of Community-Dwelling Older Adults in Taiwan"

_jpm, 2021, doi:10.3390/jpm11070653_

Round 1

Reviewer 1 Report

The Authors have carefully designed the research, but there are some shortcomings in the work. Below is a list of comments:

  1. the justification for the age of the participants is very short and is at the end of the paper. There is no information about the value of TP in younger people, not even from the literature. Assuming that the age of 60+ is the borderline age, it should be shown that TP values in the 60-64 age group will be similar to those in the under 60 group.
  2. Figure 3 - the diagram is not readable. Introducing different markers for different age groups would certainly introduce more clarity
  3. The text describes the relationship between TP and education. Therefore, it should be presented graphically
  4. The text describes how TP in men decreases with age, and the statistical differences are presented only between genders for a given age (Figure 2). Separate diagrams showing the decrease in TP in women and in men depending on age would allow the observation of the decrease in TP in a given sex.

Author Response

Dear Reviewer:

Thank you for your comments concerning our manuscript. This comment is valuable and very helpful for revising and improving our paper, as well as the important guiding significance to this research. We have revised this manuscript based on the by the reviewer’s suggestions and comments.

This manuscript has been edited by Mark Roche on Jul 07, 2021 and is considered to be improved in grammar, punctuation, spelling, verb usage, sentence structure, conciseness, general readability, writing style, and native English usage to the best of the editor's ability.

Your Sincerely,

The Authors

Reviewer 1:

The Authors have carefully designed the research, but there are some shortcomings in the work. Below is a list of comments:

  1. The justification for the age of the participants is very short and is at the end of the paper. There is no information about the value of TP in younger people, not even from the literature. Assuming that the age of 60+ is the borderline age, it should be shown that TP values in the 60-64 age group will be similar to those in the under 60 group.
  • We are grateful for the helpful suggestion. The authors had revised the introduction and justification paragraph and added the relevant tongue pressure values in the paragraph Line 122-132.
  1. Figure 3 - the diagram is not readable. Introducing different markers for different age groups would certainly introduce more clarity
  • We appreciate the constructive comments and the helpful suggestio The authors want to display the relationship between maximum tongue pressure and age of different chewing and swallowing status by scatter plots. After discussion, the authors added the graphs (Figure 4 (c) and Figure 4 (d) in Line 332 to present tongue pressure of participants without and with chewing and/or swallowing disorders by age groups.
  1. The text describes the relationship between TP and education. Therefore, it should be presented graphically
  • We are thankful for the helpful suggestio The authors added a figure (Figure 3.) to present the relationship between TP and education levels (in Line 325).
  1. The text describes how TP in men decreases with age, and the statistical differences are presented only between genders for a given age (Figure 2). Separate diagrams showing the decrease in TP in women and in men depending on age would allow the observation of the decrease in TP in a given sex.
  • We are grateful for the constructive suggestio The authors redrew the graphs (Figure 2. (a) and (b) to present maximum tongue pressure of male and female in different age groups in Line 320.

Reviewer 2 Report

This study aims to estimate tongue pressure values of adults age 60 years or older in Taiwan. The aims of this study are of value for understanding norms for tongue pressure, its relation to chewing and swallowing disorders progression, and for setting values for rehabilitative programs.

Introduction:

The introduction provides extensive detail about the anatomy and physiology of the tongue and its role in mastication and swallowing. This information could be reduced which would allow the authors to focus more on the role of tongue pressure in aging and disease progression, as well as why this information is important for personalized medicine.

The authors state that the information could be used for screening purposes (lines 146-148), but I am unsure how this information would be useful for those purposes; I would speculate that this information would be used for targeted therapy goals. Screening for functional changes in chewing and swallowing would be more appropriate (unless in the future there was evidence to show tongue pressure changes precede functional changes).

Methods:

The authors stated that “potential swallowing disorders” was a criterion for inclusion (lines 190-191). It would be helpful to have more details regarding what was considered a swallowing disorder and how this information was ascertained. Also, why were people with neurologic diseases excluded?

The authors categorized people into the chewing and swallowing disorders group (lines 260-264) based on <20 teeth, an EAT score <3, or having a swallowing disorder. How was this categorization determined and what was the rationale (particularly for including the teeth criteria)?

Why was an EAT score <3 used to indicate a chewing and swallowing disorder?

Results:

In table 1, what does “allowance” refer to? Also, what was considered a systemic disease?

In table 2, could you provide the range of tongue pressures? For individual participants, was there much variability between their three trials?

Discussion:

The authors should consider discussing studies that have presented tongue pressure values in people >60 years and if their results were similar or not. This would help to place their results in the context to the current literature.

Author Response

Dear Reviewer:

Thank you for your comments concerning our manuscript. This comment is valuable and very helpful for revising and improving our paper, as well as the important guiding significance to this research. We have revised this manuscript based on the by the reviewer’s suggestions and comments.

This manuscript has been edited by Mark Roche on Jul 07, 2021 and is considered to be improved in grammar, punctuation, spelling, verb usage, sentence structure, conciseness, general readability, writing style, and native English usage to the best of the editor's ability.

Your Sincerely,

The Authors

Reviewer 2:

Comments and Suggestions for Authors

This study aims to estimate tongue pressure values of adults age 60 years or older in Taiwan. The aims of this study are of value for understanding norms for tongue pressure, its relation to chewing and swallowing disorders progression, and for setting values for rehabilitative programs.

Introduction:

The introduction provides extensive detail about the anatomy and physiology of the tongue and its role in mastication and swallowing. This information could be reduced which would allow the authors to focus more on the role of tongue pressure in aging and disease progression, as well as why this information is important for personalized medicine.

  • We are grateful for the helpful suggestion. The authors revised the introduction paragraph in Line 133-152.

The authors state that the information could be used for screening purposes (lines 146-148), but I am unsure how this information would be useful for those purposes; I would speculate that this information would be used for targeted therapy goals. Screening for functional changes in chewing and swallowing would be more appropriate (unless in the future there was evidence to show tongue pressure changes precede functional changes).

  • We appreciate the constructive comments. As reviewer’s opinion, the TP values will be used for functional changes in chewing and swallowing. Several clinical studies supported the tongue strength exercise enabled community-dwelling elderly and stroke patients with dysphagia to increase tongue pressure significantly, to reduce the risk of airway invasion during eating and improve nutritional status. The authors made changes in Line 150-152 to fit the research purpose.

Methods:

The authors stated that “potential swallowing disorders” was a criterion for inclusion (lines 190-191). It would be helpful to have more details regarding what was considered a swallowing disorder and how this information was ascertained. Also, why were people with neurologic diseases excluded?

  • We are grateful for the helpful comments. After discussion, the authors deleted the wording of “potential swallowing disorders” in the sentence in Line 194 to avoid misunderstanding of readers. In dysphagia in people with neuromuscular and other neurological disorders, tongue dysfunction is an important factor, particularly in the oral phase. These people exhibit tongue movement problems and show tongue weakness. The authors revised the sentence in Line 194.

The authors categorized people into the chewing and swallowing disorders group (lines 260-264) based on <20 teeth, an EAT score <3, or having a swallowing disorder. How was this categorization determined and what was the rationale (particularly for including the teeth criteria)?

  • We are thankful for the comments. Total functional teeth equal 20 teeth or less is considered indicative of reducing occlusal force and chewing ability. It will indirectly affect the swallowing and decline tongue pressure. The author revised the sentence and added citation in Line 240 and Line 249-252.
  • The EAT-10 is a widely used tool designed to assess dysphagia severity in the clinical practice. A total score of 3 or more is considered indicative of dysphagia Line 219-226.

Why was an EAT score <3 used to indicate a chewing and swallowing disorder?

  • The EAT-10 is a widely used tool designed to assess dysphagia severity in the clinical practice. The normative data suggest that an EAT-10 score of 3 or higher is considered abnormal and indicates the presence of dysphagia. The authors added the sentences to describe the EAT-10 in Line 220-226.

In table 1, what does “allowance” refer to? Also, what was considered a systemic disease?

  • We appreciate the suggestio The authors corrected the wording of allowance to discretionary income and systemic disease to chronic disease in Table 1 and Table 2. (Please see in Page 12 and 13)

In table 2, could you provide the range of tongue pressures? For individual participants, was there much variability between their three trials?

  • We are thankful for the suggestio The authors have added the range of tongue pressures in Table 2. The variability among elderly closed to the results of Yamanashi et al. (https://pubmed.ncbi.nlm.nih.gov/28868806/). The participants of research by Yamanashi et al. and present study were elderly with mean age of 72.8 ± 7.4 and 73.26 ± 7.6. We thought the variability of TP was similar with previous studies.

Discussion:

The authors should consider discussing studies that have presented tongue pressure values in people >60 years and if their results were similar or not. This would help to place their results in the context to the current literature.

  • We are grateful for the helpful suggestion. The authors had revised the discussion paragraph and added the relevant tongue pressure values in Line 126-138.